# Spatial-Temporal Sensitivity Analysis of Flood Control Capability in China Based on MADM-GIS Model

**DOI:** 10.3390/e24060772

**Published:** 2022-05-30

**Authors:** Weihan Zhang, Xianghe Liu, Weihua Yu, Chenfeng Cui, Ailei Zheng

**Affiliations:** 1Key Laboratory of Agricultural Soil and Water Engineering in Arid and Semiarid Areas of Ministry of Education, Northwest A&F University, Xianyang 712100, China; zwh2660218787@gmail.com (W.Z.); liuhe710529@gmail.com (X.L.); cuichenfeng@nwafu.edu.cn (C.C.); 2College of Water Resources and Architecture Engineering, Northwest A&F University, Xianyang 712100, China; zal@nwafu.edu.cn

**Keywords:** MADM-GIS model, 3D-TOPSIS method, ROC curve, visualization of flood control capacity, entropy weight method

## Abstract

To facilitate better implementation of flood control and risk mitigation strategies, a model for evaluating the flood defense capability of China is proposed in this study. First, nine indicators such as slope and precipitation intensity are extracted from four aspects: objective inclusiveness, subjective prevention, etc. Secondly, the entropy weight method in the multi-attribute decision making (MADM) model and the improved three-dimensional technique for order preference by similarity to ideal solution (3D-TOPSIS) method were combined to construct a flood defense capacity index evaluation system. Finally, the receiver operating characteristic (ROC) curve and the Taylor plot method were innovatively used to test the model and indicators. The results show that nationwide, there is fine flood defense performance in Shandong, Jiangsu and room for improvement in Guangxi, Chongqing, Tibet and Qinghai. The good representativity of nine indicators selected by the model was verified by the Taylor plot. Simultaneously, the ROC calculated area under the curve (AUC) was 70%, which proved the good problem-solving ability of the MADM-GIS model. An accurate assessment of the sensitivity of flood control capacity in China was achieved, and it is suitable for situations where data is scarce or discontinuous. It provided scientific reference value for the planning and implementation of China’s flood defense and disaster reduction projects and emergency safety strategies.

## 1. Introduction

Promoted by the need for economic development, the global water cycle has accelerated substantially, causing a series of abnormal climate changes [1,2]. According to data released by the Ministry of Water Resources of China, the areas affected by flood disasters in southern China in 2020 included 198 rivers in 27 districts across the country, with a total of 30.2 million people involved, and a loss of CNY 61.79 billion as of 14:00 on 9 July 2020. The healthy and stable development of China’s economy and society [3,4] has been hindered by the expansion in the scope and degree of extreme climate events. Therefore, the establishment of a flood control capability evaluation model is crucial for maintaining infrastructure such as reservoirs, reasonably managing and controlling water resources, and effectively protecting the lives and properties of the Chinese people. It is increasingly necessary to assess the flood disaster defense capability in urban hydrology.

As a common natural disaster, urban floods generate multitudes of negative impacts on people across the country. Multi-level studies on the evaluation of flood defense capacity have been carried out worldwide. The main reasons for flood disasters are unreasonable urban planning and uncoordinated flood management systems. The emergence of excess runoff is caused by short-term intense rainfall that exceeds the capacity of the drainage system [5,6], which reflects the deep relationship between flood disasters and geographical, social and other factors. Studies reveal that urban waterlogging is mainly attributable to dual aspects [7], including (1) a natural perspective: global warming indirectly increases the probability of urban rainstorms, and (2) a social standpoint: it reflects the impact on urbanization and the urban water cycle. There are many approaches that have been developed for the evaluation of pluvial flooding. The related research [8] mainly revolves around the above two aspects, including six factors, such as meteorological factors and urbanization factors. Additionally, Elmer, F. et al. [9] showed that the increase in direct flood damage observed over the last decades may have been caused by changes in the meteorological drivers of floods, or by socioeconomic development. At the same time, varieties of urban waterlogging caused by climate change have a significant influence on economic growth worldwide [10]. Economic factors are not only one of the causes of urban floods, but also passively affected by the occurrence of disasters. So, the forecasting task for urban development in flood-prone areas will be further complicated because of the interaction between the two factors [11]. Basically, the most general type of forecasting method is based on GIS and remote sensing (RS) technology to draw flood maps [12,13,14,15,16,17,18,19,20]. The flood disasters are monitored based on the information from different RS platforms and bands, and the possibility of applying composite information to monitor potential flooding is analyzed. The disadvantage of this method is that the investment in building and maintaining a platform is heavier than that in the theoretical research method. Another approach is a comprehensive weighting method [21,22,23,24] that can more comprehensively solve the problem by constructing an index system. However, there is no fixed system; if the indicators of different zoning systems vary, there will be extreme contradictions between the results of the risk assessment and the conclusions drawn. Research studies have reportedly evaluated the flood defense capacity of a specific area [25] by using a combination of the entropy weight method and the analytic hierarchy process (AHP) method [26,27,28,29,30] or the individual TOPSIS method [31]. As a hierarchical multi-objective comprehensive evaluation method, AHP plays a good evaluation role when the evaluated object’s attributes are difficult to quantify. However, it is incredibly subjective to rely on the weight of expert scoring. The problem is that there are too many decision-making layers for evaluation, and the gap between the judgment matrix and the consensus matrix is too large to use AHP. This is well solved by TOPSIS. The weight is wholly derived from the data, and the objectivity is better than that of AHP. However, the degree of data dependence is deep, and some of the indicators are not always adaptable to the information, which will harm the model’s accuracy.

Various research studies have been carried out by scientists in other parts of the world to approach the topic. Zhengzhou, a city in southern China, was selected as the study area by Lin, Lin et al. to build a flood-susceptibility map, which was generated by using GIS spatial analysis tools and the analytic hierarchy process method [32]. Wang, Yamei et al. used a semi-quantitative model and fuzzy analytic hierarchy process (FAHP) weighting approach to assessing flood risk in the Dongting Lake region of Hunan Province, which is in central China [33]. The results of this article were compared with the above two articles, expanding the study area. Additionally, some articles published by Chun, Xiang et al. [34,35,36,37] combined the theoretical method of analysis with actual flood control capacity evaluations, and analyzed the relevant influencing factors. They presented relevant theoretical frameworks, but many lacked accurate data support, were not experimentally validated, and could only be used as theoretical references.

This study presents a regional flood defense capacity evaluation model, called MADM-GIS, to facilitate the development of flood mitigation strategies and better flood control. It is the first attempt to treat all regions of China as the research area in the field and over a time range that is more extended than other papers for a more comprehensive study. According to a survey of the existing literature, a richer indicator evaluation system than that of Shuqi, Wu et al. [38,39] was created, and the single indicator problem in the traditional 2D-TOPSIS was avoided by using the improved 3D-TOPSIS method in this paper [40]. The entropy weight method was also creatively combined with multiple spatial analysis tools in the ArcGIS platform [41,42,43] to evaluate the flood control capability comprehensively and realize the visualization of data. The ROC curve [44] was firstly used to test the flood defense capability evaluation model, and the Taylor diagram [45,46,47] was innovatively applied to test the degree of representation of the nine indicators. The obtained verification results were satisfactory.

## 2. Study Area and Data Resources

### 2.1. Study Area

China is hit by heavy precipitation and flood disasters almost every year, so this paper targets the whole of China as the research area for the first time in the flood defense capability assessment. China’s longitude ranges from 73°33′ E to 135°05′ E, and its latitude ranges from 3°51′ N to 53°33′ N. The terrain is high in the west and low in the east, forming three steps from west to east. The land area is about 9.6 million square kilometers, of which mountains, plateaus and hills account for about 67%. At present, China has 34 provincial-level administrative regions, including 23 provinces, 5 autonomous territories, 4 municipalities directly under the Central Government, and 2 particular administrative regions. Due to the lack of data on Hong Kong, Macau and Taiwan, this article will not discuss them.

China’s climate is complex and diverse, spanning five climatic zones from south to north. According to the 2020 "China Climate Bulletin", a total of 37 nationwide regional rainstorms occurred in 2020, and the annual number of rainstorm days since 1961 was second only to 2016. The Yangtze River is the longest in Asia, and the Yellow River is the second longest river in China. In the summer of 2020, they received the highest precipitation since 1961, while the second-highest was recorded in the Huaihe River and Lake Taihu during the same period. In 2020, the country’s total annual precipitation resources measured 6592.65 billion cubic meters, 616.33 billion more than average.

### 2.2. Data Resources

Among the evaluation indicators related to flood defense capability that were selected in this paper, the regional green area, the population density, the number of medical institutions, GDP, and related data on the flood control area were obtained from the annual data for each province in China’s National Statistical Yearbook.

DEM, the spatial resolution of the precipitation intensity data, is 1:1,000,000 from the National Basic Geographic Information Center in China. The fragmented data were implemented in accordance with the GB/T 13989-2012 “National Basic Scale Topographic Map Framing and Numbering”. The space storage unit was 6° (longitude difference) × 4° (latitude difference). The slope data was obtained by surface analysis in the spatial analysis tool of ArcGIS based on DEM data. The main river data’s spatial resolution was 1:4,000,000, which came from the National Basic Geographic Information System Vector Data. It needed to be converted into raster format by using the conversion tool in ArcGIS. The raster data on river network density could be obtained through the line density analysis in the spatial analysis tool. They were converted into a scale of 1:1,000,000 through the resampling extension module in ArcGIS to ensure that the raster calculation was performed based on the same spatial resolution. Otherwise, the subsequent algebraic calculation of raster data would be meaningless. The objective flood data in the final inspection process were from the Global Disaster Data Platform. The return period for a general flood is 10 years; for a large flood, 10–20 years; and for a major flood, 20–50 years. Thus, in this paper, we selected the data for various indicators in China from 2001 to 2020. Abbreviations for every Chinese region are shown in Table 1. 

## 3. Methodology

### 3.1. Analysis Framework

The new flood control capability analysis framework is mainly divided into four parts. The detailed content will be introduced in terms of the following points:(1)Data collection and processing;(2)Flood control capability evaluation index system;(3)Flood control capability evaluation and calculation;(4)Model validation.

There is a progressive relationship between the above four parts, forming a complete evaluation system. As long as the relevant required data are input, a systematic evaluation of the flood control capacity of any region can be realized. At the same time, the corresponding parameters can be collectively analyzed and adjusted according to the specific situation, and the comparative experiments can be carried out.

Every five years is regarded as one research period; the 20 years span was divided into a total of four periods: 2001–2005, 2006–1010, 2011–2015, and 2016–2020. Each stage generated an evaluation map, in line with the speed of the strategic policy of China’s five-year plan and social development. Then, this paper compared the changes in flood sensitivity indicators in each period to profoundly and systematically explore the impact of each indicator on the flood defense capabilities in various regions. It was shown a brief overview of the article writing process by Figure 1.

### 3.2. Flood Control Capability Evaluation Index System

Appropriate parameter indicators play a decisive role in accurately assessing the scale of floods and implementing universal defense measures. Based on a reading of the existing literature [48,49], this paper divided the indicators into four aspects:(1)$$FCS=f\left(OIF,SPF,FMP,FHD\right)$$
where FCS denotes the evaluation results of the MADM-GIS model regarding flood defense capability. OIF refers to the region’s geographical and meteorological conditions and other natural environmental aspects that can tolerate rainstorms and floods and maintain a stable state; it can reflect the macro self-regulation ability of the natural environment, thereby reducing the frequency of extreme disasters. SPF refers to the ability of humans to forecast, prevent and defend against the disasters; it reflects the nature of the ability of the Chinese government and relevant departments in various regions to respond to flood disasters promptly, protect the safety of people’s lives and their properties across the country, and minimize the loss.FMP refers to the ability of human beings to participate in economic, medical, social, and other activities to reduce the negative impact of disasters under the condition that natural conditions cannot change. FHD refers to the nature of the intensity of the direct factors leading to floods. Nine indicators, including DEM, slope, precipitation intensity, gross regional product, number of medical institutions, urban population density, urban green space area, and flood control area [50,51], were extracted from four aspects to form a flood defense capacity evaluation system. Table 2 is the list of factor abbreviations and acronyms.

### 3.3. MADM-GIS Model

Multi-criteria decision-making (MCDM) is a decision to choose among a finite (infinite) set of conflicting and incommensurable schemes. Its origins [52] can be traced back to the concept of Pareto optimization (1896), and it was brought into the decision-making field as a normative decision-making method in the 1960s, represented by the research on objective planning by Charnes and Cooper. MCDM methods are based on the principle of proposing the best solution among the schemes under certain criteria, so they have been used more widely recently [53]. There was a Monte Carlo Simulation used in a multicriteria decision model [54] aiming to prioritize flood risks in urban areas under climate effects. They also have been applied in many specific fields to solve some selection and ranking problems, such as information technology [55], design and development [56], civil engineering and management [57], renewable energy [58] and medical diagnosis [59]. There are more than 100 MCDM methods, and each of them has its own performance capacity and characteristics, which are often associated with the model’s computational process and methodology.

MCDM is mainly divided into two categories: multi-attribute decision-making (MADM) and multiple objective decision-making (MODM), according to whether the decision-making scheme is limited or unlimited. The MCDM model has been used to evaluate flood defense capacity in the existing literature [33,60]. Compared with MADM, the scope of application of MCDM is broader, but inaccurate evaluation results are usually generated. The difficulty associated with ineffective flood control capability evaluation caused by an extensive data range processed by MCDM has been solved.

In this paper, the evaluation method for the index system was innovatively combined with the flood control capability evaluation; the two methods, including the TOPSIS method and the entropy weight method under MADM, were chosen, and MATLAB was utilized for the calculations. GIS was used to generate the distribution map of the corresponding period and location of each indicator proposed by the MADM-GIS model. As it is the first of its kind built in this field, with efficient and accurate assessment simultaneously completed, this model represents a relatively new attempt with a certain degree of practical value. Figure 2 is the MCDM logic diagram that shows the classification and content of MCDM.

#### 3.3.1. The Entropy Weight Method

The entropy weight method is an objective weighting method based on the degree of variation in the index. The lower the amount of information reflected by the index, the lower the corresponding weight will be. Table 3 is the key math symbols and their meanings.

A positive matrix is established according to the data, which is convenient for subsequent data preprocessing:(2)$$\;\;\;P=\left[\begin{array}{ccc} a_{1,1} & \cdots & a_{1,9}\\ \vdots & \ddots & \vdots \\ a_{31,1} & \cdots & a_{31,1} \end{array}\right]$$

The most common standardization method, known as the dispersion standardization method, was used in this paper to transform the original data linearly. The results were mapped to the [0, 1] interval, eliminating the interference of different units and incomparability between indicators.
(3)$$\tilde{Z_{ij}}=\frac{a_{ij}-min\left\{a_{1j},a_{2j}\ldots a_{nj}\right\}}{max\left\{a_{1j},a_{2j}\ldots a_{nj}\right\}-min\left\{a_{1j},a_{2j}\ldots a_{nj}\right\}}$$

The proportion of the *i*-th sample under the *j*-th index is calculated, and it is regarded as the probability calculation probability matrix *F* used in the relative entropy calculation, where the calculation formula of $F_{ij}$ is:(4)$$F_{ij}=\frac{a_{ij}}{\sum_{i=1}^{n}a_{ij}}$$

The greater the information entropy of the event is, the smaller the amount of existing information will be. Furthermore, the amount of information that can be supplemented will be larger.

For the *j*-th index, the calculation formula of its information entropy is:(5)$$s_{ij}=-\frac{1}{\ln n}\overset{31}{\underset{i=1}{\sum }}F_{ij}\log(F_{ij})$$

The information utility value is defined as:(6)$$\begin{array}{c} \;\;\;\;\;m_{j}=1-s_{j} \end{array}$$

The weight of each indicator can be obtained by normalizing the information utility value:(7)$$W_{j}=\frac{m_{j}}{\sum_{j=1}^{m}m_{j}}$$

In the process of solving the MADM problem, the weight of the attribute is used to reflect the relative importance of the attribute, which plays a pivotal role. In this paper, the entropy weight method was used to calculate the respective weights of the nine indicators on the impact of flood defense capacity. Figure 3 shows that the weight of P changed abruptly from the first period to the second period and was relatively stable in the following 15 years, following a downward trend. However, the process of change in M and P was precisely the opposite. There was a sudden increase in the transition from the first five years to the second, and then there was a slight increase. Only the weight of urban green space decreased gradually and uniformly over time; the corresponding graphs of the six indicators W, D, C, S, R, and G in the radar chart all transitioned from inside to outside, which gradually increased over time.

#### 3.3.2. 3D-TOPSIS Model

The traditional TOPSIS is based on two-dimensional indicators. Only a score for each indicator individually can be obtained, which is inconvenient for visualization. The two-dimensional model was improved to three-dimensional in this paper, and the total score for flood defense capacity in every region was obtained by superimposition. Table 4 is the key math symbols and their meanings.

The negative indicators need to be converted into positive indicators, including R, D, S, C, and the transformed data matrix is still recorded as *P*.
(8)$$\;q_{ij}=q_{max}-\dot{q_{ij}}$$

The original data should be normalized to avoid large network prediction errors due to the large magnitude difference between input and output data:(9)$$\;\;\;T_{ij}=\frac{q_{ij}}{\sqrt{\sum_{i=1}^{n}q_{ij}{}^{2}}}$$

Based on the weights obtained above, a standardization matrix is constructed:(10)$$Z=\left[\begin{array}{ccc} q_{1\;1}\cdot \omega_{1} & \cdots & q_{1\;9}\cdot \omega_{1}\\ \vdots & \ddots & \vdots \\ q_{31\;1}\cdot \omega_{31} & \cdots & q_{31\;9}\cdot \omega_{31} \end{array}\right]$$

The maximum and minimum values are defined as:(11)$$\begin{array}{c} \;\;\;\;\;\;\;\;\;\;Q^{+}=\left(max\left\{a_{1\;1},a_{2\;1}\ldots a_{n\;1}\right\},max\left\{a_{1\;2},a_{2\;2}\ldots a_{n\;2}\right\},\ldots ,max\left\{a_{1\;9},a_{2\;9}\ldots a_{n\;9}\right\}\right)\\ \;\;\;\;\;\;\;Q^{-}=\left(min\left\{a_{1\;1},a_{2\;1}\ldots a_{n\;1}\right\},min\left\{a_{1\;2},a_{2\;2}\ldots a_{n\;2}\right\},\ldots ,min\left\{a_{1\;9},a_{2\;9}\ldots a_{n\;9}\right\}\right)\# \end{array}$$

The distances between the maximum and minimum values of the evaluation object *i*-th (*i* = 1, 2, … 31) are respectively defined as:(12)$$\begin{array}{c} \;\;\;\;\;\;\;\;\;\;\;\;\;\;\;\;\;\;\;\;\;\;\;\;\;\;\;\;\;\;\;\;\;\;\;\;\;\;\;\;\;\;\;\;\;\;\;\;\;\;\;\;\;\;D_{i}{}^{+}=\sqrt{\overset{m}{\underset{j=1}{\sum }}{\left(Q_{j}{}^{+}-q_{ij}\right)}^{2}}\\ \;\;\;\;\;\;\;\;\;\;\;\;\;\;\;\;\;\;\;\;\;\;\;\;\;\;\;\;\;\;\;\;\;\;\;\;\;\;\;\;\;\;\;\;\;\;\;\;\;\;\;\;\;D_{i}{}^{-}=\sqrt{\overset{m}{\underset{j=1}{\sum }}{\left(Q_{j}{}^{-}-q_{ij}\right)}^{2}}\# \end{array}$$

### 3.4. Taylor Diagram

The Taylor plots in the existing literature were used to test extreme air temperature, the meteorological conditions in different models, or the correlation analysis of correlation coefficients (CC). It is a new type of ternary diagram that was firstly introduced in the studied field, which is about the evaluation of flood defense capability. In the figure, the model is represented by scattering points, the correlation coefficient (CC) is shown by the radial line, the standard deviation (SD) is represented by the horizontal and vertical axes, and the root mean square error (RMSE) is indicated by the dotted line. The CC refers to similarity between the simulation results of different indicators and the observed values. The differences between the spatial uniformity of simulation results of the models and the observations are respectively reflected by RMSE and SD. The smaller the SD is, the more stable the model is.

The results are as follows. The SD is 0.2–0.4, which is in line with the characteristics of the heterogeneous coverage of the nine indicators, and the points of data are clustered around the mean to a small degree. The correlation between the nine indicators is weak, which is shown by the CC as 0.1–0.9. The combination of the two axes shows that the nine indicators are highly representative of the research problem, and the problem of the high degree of index coincidence reducing the model’s accuracy can be solved. The deviation between the observed and actual values measured by the RMSE is 0.2–0.5, which indicates that the departure of the data in this paper was slight and the applicable standard was met. The test results show that the established MADM-GIS model and the weights corresponding to the indicators are close to the research problem and have practical significance and scientific and objective persuasion. Figure 4 is the Taylor plot of nine indicators from 2001 to 2020.

## 4. Results and Discussion

### 4.1. Analysis and Validation

The time interval of the data is from 2001 to 2020, which is divided into four stages, showing the trend of influencing factors over time in detail. Nine indicators extracted from four aspects were shown in Table 1; from the perspective of time series, they are divided into two parts including six indicators from a timing perspective that change significantly with time U, W, P, R, G, M and D, S, C that almost don’t change. The data is imported into ArcGIS, and divided into 8 categories by the natural breakpoint method. The higher the level of the region is, the better the performance of the corresponding index is, and the corresponding province shows a darker color.

### 4.2. 3D-TOPSIS Model Parameter Analysis

#### 4.2.1. Urban Green Area

Figure 5 is the spatial-temporal distribution of urban green space in China. From Figure 5a, the regional differences in urban green space area from 2001 to 2005 are the most obvious. The urban green areas of the regions such as Sichuan, Guangdong, Jiangsu, Shanghai are outstanding in China. Comparison of the four maps shows that the growth rate of the value in Qinghai and Tibet was relatively slow because western China had a higher altitude, which is not conducive to vegetation growth. From Figure 5a–d, the differences of the regions were gradually decreasing. The value in Inner Mongolia and northeast China increased significantly and then tend to be stable, which indicates that the green engineering in China has been better developed with significant results under the advancement of China’s five-year plan.

#### 4.2.2. Waterlogging Prevention Area

The area of waterlogging removal refers to an area where waterlogging-prone farmland is rescued from flooding due to the construction of waterlogging control projects or other water conservancy facilities, and the standard is reached more than once in three years. It reflects the protection and utilization of water resources in China. Figure 6 is the spatial-temporal distribution of waterlogging prevention and control areas in China. From Figure 6a, the flood defense in Inner Mongolia, Tibet, Sichuan and Shandong increased significantly from 2001 to 2005. In Figure 6b, flood defense in southeastern China continued to increase within a small range. The work performed by Guangdong, Shandong, Henan, and Sichuan was excellent. To sum up, as shown in Figure 6d, the efficiency of releasing policies for flood defense and disaster reduction across the country gradually accelerated. The area of waterlogging prevention reached a balanced level, and rational water resources management was comprehensively promoted.

#### 4.2.3. Population Density

Urban population density refers to the number of people per unit area. Figure 7 is the spatial-temporal distribution of population density in China. As vividly shown in Figure 7a, owing to the inferior backdrop of development, the population density in Xinjiang, Heilongjiang, Inner Mongolia, Sichuan, Yunnan and Guangxi is very sparse. It shows that the period of 2006–2010 was at the peak of regional differences in Figure 7b,c with the implementation of the five-year plan. In Figure 7b, population density in Xinjiang and Heilongjiang increased significantly, and also increased widely in southern China. With the continuous development of China’s economy and society, the population problem had been solved, and the density of the urban population was increasing nationwide. The regional differences in terms of population density across the country were becoming increasingly slim, which is graphically depicted by Figure 7d. Due to the inception of the two-child policy, the population density across the country has basically become saturated.

#### 4.2.4. Medical Institutions

The number of medical institutions refers to the total number of health institutions established in accordance with legal procedures that engage in disease diagnosis and treatment activities. Figure 8 is the spatial-temporal changes in the number of medical institutions in China from 2001 to 2020. It represents the number of medical and health resources and reflects the ability of a region to provide medical services when floods come. The higher the flood control ability score climbs, the better the medical configuration is. There will be more medical resources that can be used to ensure the safety of people’s lives and property. During 2001–2005, Sichuan, Hunan, and Guangdong’s numbers of medical institutions were among the top three nationwide. In the second period, Beijing, Hebei, and Henan performed more outstandingly in this regard, and the number of medical institutions increased significantly. 

#### 4.2.5. GDP

Gross domestic product (GDP) is the core indicator of national economic accounting, which can measure the financial status and development level of a country or region and represents the financial ability to deal with floods. Figure 9 is the chart of temporal and spatial changes in China’s GDP. The higher GDP is, the more investment that can be allocated to flood recovery. Meanwhile, the damage caused by a flood can be quickly mitigated to a great extent. From 2001 to2005, Jiangsu, Shandong, Guangdong, Zhejiang, and Shanghai were in the leading position in China. Due to the impact of China’s five-year plan on the economy, economic development improved steadily in Beijing, Hebei, Henan and Sichuan from 2006 to 2010. The northern regions also responded to the call, which made a big difference. From the graph, the southern region’s GDP improved in an all-around way. People’s living standards and social development levels were promoted under the economic drive.

#### 4.2.6. Rainfall Intensity

Rainfall intensity refers to the amount of rainfall in a unit of time. Figure 10 is the spatial-temporal distribution of rainfall intensity in China from 2001 to 2020. It can be seen from the four figures that the spatial distribution of rainfall intensity in China is strong in the south and weak in the north, and there is an apparent planar aggregation phenomenon in the rainfall in the south. In Figure 10a, the rainfall intensity was the strongest in the four periods. The rainfall intensity was lower than the previous five years in Figure 10b. Yunnan, Sichuan, Guangxi, Hunan, Jiangxi and Chongqing are prominent gathering centers. The transition process from the four figures shows that the aggregation phenomenon was becoming more and more apparent, and a point-like aggregation can be seen in Figure 10c. Hunan, Guangxi, Jiangxi, Guangdong, and Fujian were the main centers of the point aggregation phenomenon. Close attention to each gathering center should be encouraged, and the management of extreme precipitation events should be strengthened. The efficiency of monitoring and forecasting needs to be improved. The government should provide exceptional policy support, and the surrounding provinces should also provide assistance.

#### 4.2.7. DEM, Slope and Drainage Density

The steeper the slope is, the more serious the soil erosion phenomenon will be. The surface runoff and infiltration will be affected. Floods will be caused by the surface runoff to a certain extent, and the DEM determines the slope. Due to the regulation and storage of the basin, the floods are in the form of fluctuations. According to the law of conservation of energy, the higher the DEM is, the greater the potential energy of the water flow. In the process of downward flow, part of the potential energy of the flood is converted into kinetic energy, and the flow velocity is accelerated, increasing the risk that the resulting severe shock will cause harm downstream. Figure 11 is the spatial distribution map of DEM. Figure 12 is the spatial distribution map of slope. Figure 13 is the spatial distribution map of major river networks.

### 4.3. MADM-GIS Model

#### 4.3.1. Analysis of Results of Entropy Weight Method 

The results show that the weight of P changed abruptly from the first period to the second period and was relatively stable in the following 15 years, showing a downward trend. However, the process of change in M and P is precisely the opposite. There is a sudden increase in the transition from the first five years to the second, and then there is a slight increase. Only the weight of urban green space decreases gradually and uniformly over time; the corresponding graphs of the six indicators W, D, C, S, R, and G in the radar chart all transition from inside to outside, which gradually increases over time.

#### 4.3.2. Analysis of Results of D-TOPSIS Model

$S_{i}$ is defined as the relative proximity, and the calculation formula is:(13)$$S_{i}=\frac{D_{i}{}^{-}}{D_{i}{}^{+}+D_{i}{}^{-}}$$

The larger the obtained evaluation matrix *S_i_* is, the greater the degree of closeness to the optimal solution, and the flood defense capability of the area is more robust. When $S_{i}=0$, it is the lowest flood control capability; $S_{i}=1$ is the highest. Finally, sort according to the size of the proximity. The larger the value is, the closer the evaluation object to the optimal solution is.

The weights in 2001–2005 are taken as an example:(14)$$OIF=\overset{2}{\underset{i=1}{\sum }}x_{i}\omega_{i}=0.15063\times U+0.03942\times C\;\;FMP=\overset{2}{\underset{i=1}{\sum }}x_{i}\omega_{i}=0.06064\times M+0.09591\times G\;SPF=\overset{2}{\underset{i=1}{\sum }}x_{i}\omega_{i}=0.21922\times W+0.11295\times P\;FHD=\overset{2}{\underset{i=1}{\sum }}x_{i}\omega_{i}=0.07479\times R+0.17401\times D+0.17143\times S\;FCS=OIF+FMP+SPF+FHD$$

To obtain the flood defense capacity index for each period, the linear weighted sum method (LWSM) was used to perform statistical processing on the data. The LWSM is an evaluation function method, which can solve multi-objective programming problems by assigning corresponding weight coefficients to each objective and then optimizing its linear combination. Each basic indicator was multiplied by the corresponding weight and then summed up. Multiple base metrics were converted into a single numerical index that was regarded as the final score. Through the value of the final score, it was possible to quantify and compare the flood defense capability of each city.

From Figure 14, it can be seen that the evaluation scores and absolute values of the direct economic losses in the 31 provinces show roughly the same trend. As is shown, the results of the model solution are in line with objective reality, which verifies the validity of the model. However, the evaluation score does not match the absolute value of the direct economic loss in Sichuan, Shaanxi, Jiangxi and Hunan. It was found that the occurrence of this phenomenon is strongly related with social factors outside the model that did not belong within the scope of the research through actual investigation. There was no effect on the analysis result, namely, that the higher the score of the MADM-GIS model is, the lower the value of the direct loss rate caused by the flood is. The evaluation results for regional flood defense capacity are only related to the actual data, and the government staff in the relevant regions need to formulate policies and implement various adjustments.

Figure 15a is about the flood defense capability assessment scores in China obtained with the MADM-GIS model. Figure 15b is about the direct losses that are caused by floods and is based on actual data in China. The stronger the flood defense capability of a region is, the less the direct loss is, which is in line with objective reality. The two variables of flood defense capacity and direct loss in Xinjiang, Gansu, and Inner Mongolia are all at the middle level in China. There is little difference in the comparison of two figures, whose colors are roughly the same. Tibet, Yunnan, Guizhou, Chongqing, and Guangxi are mainly located in western China, where flood defense capacity is lower, and direct losses are heavier than in other regions. The eastern regions, including Hebei, Henan, Jiangsu, Shanghai, Guangdong, and Zhejiang, played an exemplary role with higher evaluation scores in coping with severe floods.

### 4.4. MADM-GIS Model Validation

Analysis of ROC Curve Results 

Figure 16 is the ROC curve of the flood control capacity score to the MADM-GIS model. The sensitivity and specificity of variables are reflected by the ROC curve [61,62], and intuitive comparison of different test methods is provided under the same scale. It was drawn by SPSS software to test the accuracy and feasibility of the model. There are two advantages of this method. First, there is a broader application range to indirectly analyze various types of raw data. Secondly, the systematic analysis of multiple covariates can be carried out, which is more advanced than the ROC curve in analyzing univariate raw data described in the current literature [50,51,63].

The closer the inflection point is to the upper left corner of the figure, the greater the diagnostic value of the model is. The accuracy of test objects was indicated by the AUC of 0.7 to 0.9; the closer the AUC is to 1, the better the diagnostic effect of the model. As is shown, the AUC was 70%, and the accuracy and feasibility of the MADM-GIS model were meaningful.

## 5. Conclusions

MADM-GIS, a flood defense capability evaluation model with strong inclusiveness and adaptability, was highlighted in this paper. It is the first attempt to evaluate the flood defense capability in every region of China, and the problem of China’s poor response to flooding disasters in the past was resolved. First, the data visualization platform based on ArcGIS was assisted by MATLAB and SPSS software. It was established to display the time-space pattern of the calculation results of the indicators and analyze the process of change in the nine indicators during the past 20 years. Combined with the entropy weight method and the improved 3D-TOPSIS method, the defense capability scores of various regions were obtained, and the corresponding spatiotemporal distribution map was generated. Then, the ROC curve was utilized as the model to test the method. As to the AUC = 70%, it was a satisfactory result, and the Taylor diagram was combined to intuitively represent enough adaptation of each index. Finally, the evaluation ability under different index modes was compared, and the usability and accuracy of the MADM-GIS model were verified.

The results of analysis show that the main influencing factors in the evaluation of flood defense capacity were W, U, and D, and that their weights hardly changed with time, which indicated they were the top three of the nine indicators. From 2001 to 2020, the differences in urban green space between regions were gradually narrowing, and there were upstanding effects in waterlogging control displayed in Guangdong, Shandong, Henan, and Sichuan. GDP in the southern region increased significantly, and the capacity of medical resources support for rainstorm and flood disasters was strengthened. From 2016 to 2020, the regional differences in overall population density across the country gradually decreased, and the population density was nearly saturated. The most obvious point-like aggregation phenomenon in regional rainfall intensity was shown from 2011 to 2015, and Hunan, Guangxi, Jiangxi, Guangdong, and Fujian were mainly at the center of the point-like aggregation. The final evaluation results show that the flood control capacity of Tibet, Yunnan, Guizhou, Chongqing and Guangxi needs to be improved significantly, and there is a plenty of room for improvement. Hebei, Henan, Jiangsu, Shanghai, Guangdong, and Zhejiang are in the leading positions in China.

The spatial situation for flood defense capabilities in China was described in this paper and will contribute to enriching the system for evaluation of urban flood control capabilities. An objective and scientific reference plan was provided for the measures taken by the Ministry of Water Resources and related departments to deal with flood control and disaster mitigation.

Since the research scope covered the whole area of China, only the main factors were selected for the evaluation indicators to prevent interference from secondary factors with the research conclusions of the model. Therefore, there were regional limitations in the promotion and application of the model. For the evaluation of flood defense capacity in smaller areas, the application of the model needs to be combined with the actual situation in each region to obtain more accurate and reliable research results with detailed evaluation indicators. This paper selected the data from 2001 to 2020, which were acquired in combination with data on China’s political development and economic development. For more targeted research, the time range can be expanded and the database can be enriched to analyze more factors that contribute to the results. More indicators were selected in this paper than for ordinary evaluation models; hence, there was no detailed analysis of the sensitivity between indicators. Follow-up data and further experiments are needed to enrich the contents of the model and improve its efficiency.

## Figures and Tables

**Figure 1 entropy-24-00772-f001:**
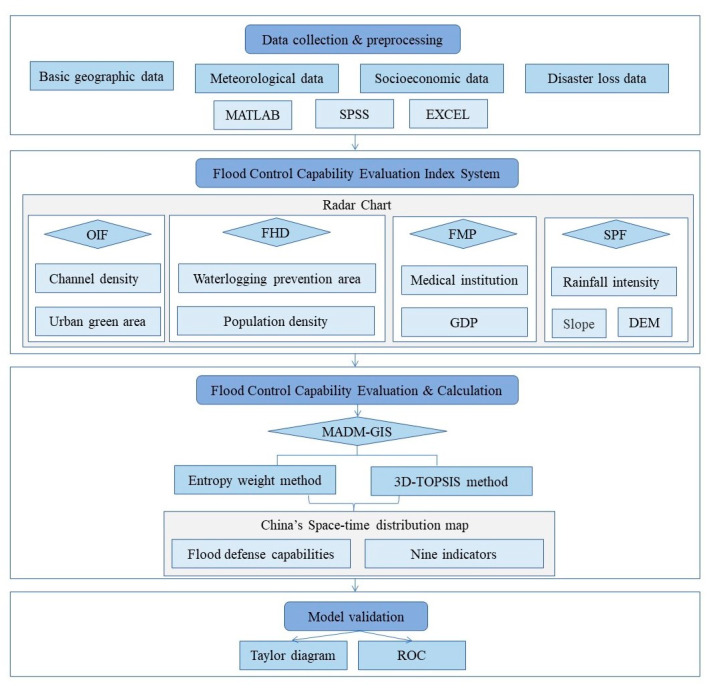
Flow chart for building the model for evaluation of China’s flood defense capability.

**Figure 2 entropy-24-00772-f002:**
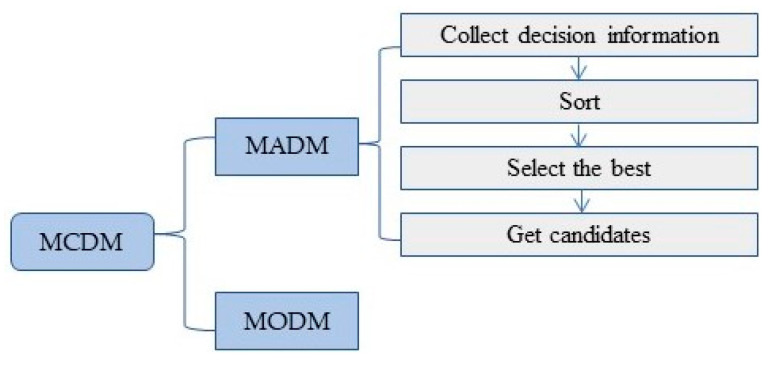
MCDM logic diagram.

**Figure 3 entropy-24-00772-f003:**
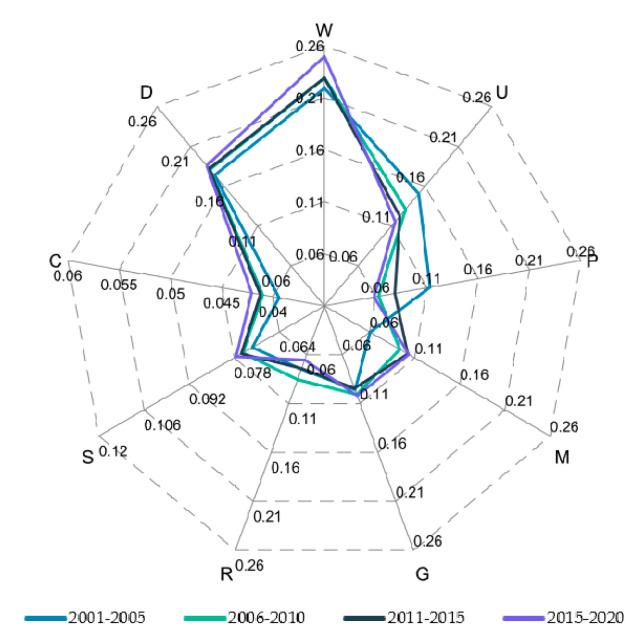
Nine indicator weight radar diagram.

**Figure 4 entropy-24-00772-f004:**
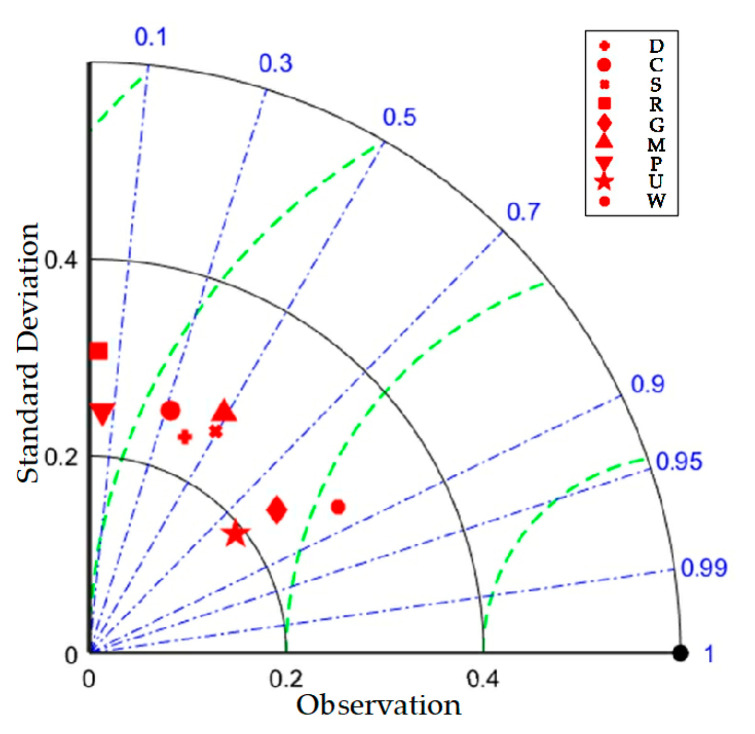
Taylor plot of nine indicators from 2001 to 2020.

**Figure 5 entropy-24-00772-f005:**
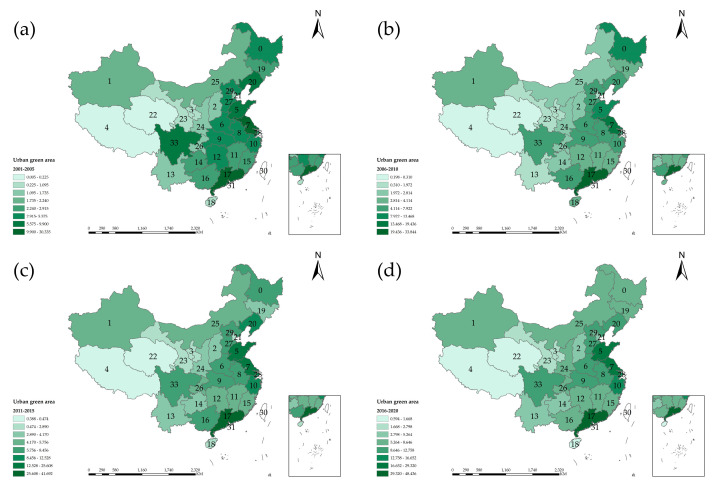
Spatial and temporal distribution of urban green space in China.

**Figure 6 entropy-24-00772-f006:**
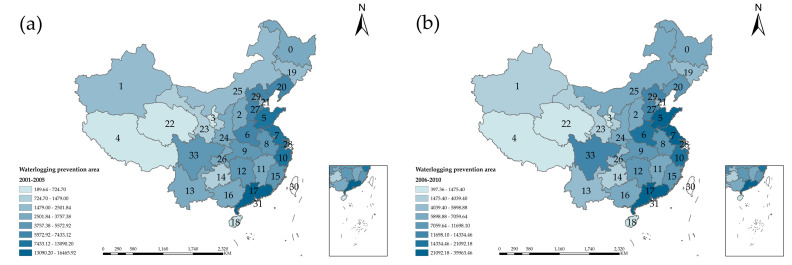
Spatial and temporal distribution of waterlogging prevention and control areas in China.

**Figure 7 entropy-24-00772-f007:**
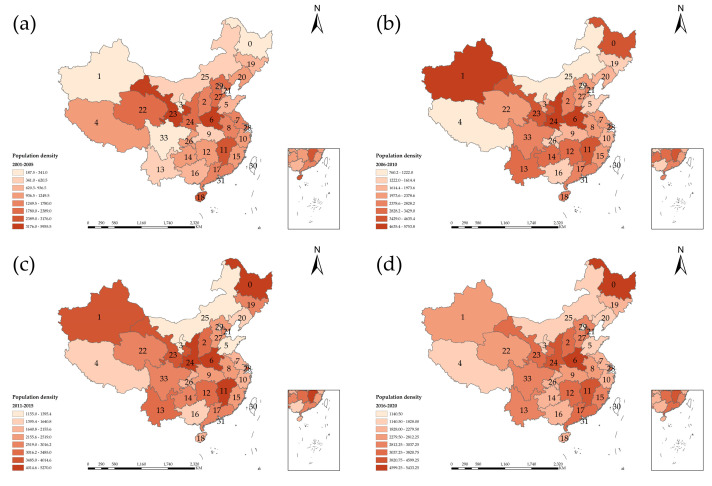
Spatial and temporal distribution of population density in China.

**Figure 8 entropy-24-00772-f008:**
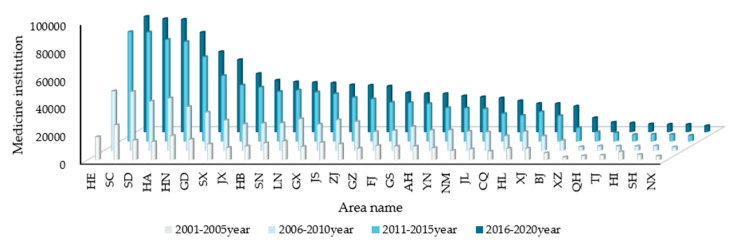
Spatial and temporal changes in the number of medical institutions in China from 2001 to 2020.

**Figure 9 entropy-24-00772-f009:**
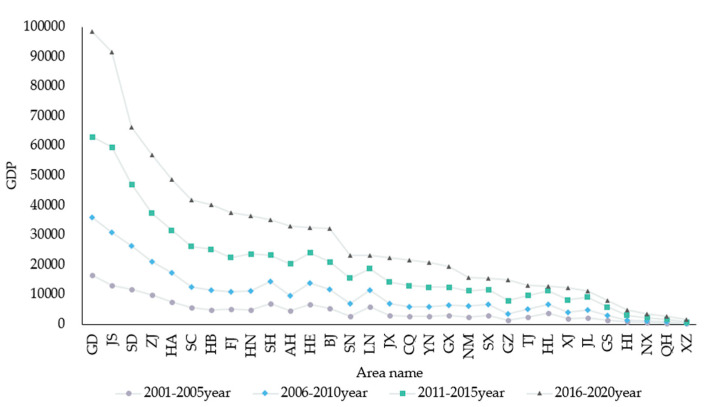
Chart of temporal and spatial changes in China’s GDP.

**Figure 10 entropy-24-00772-f010:**
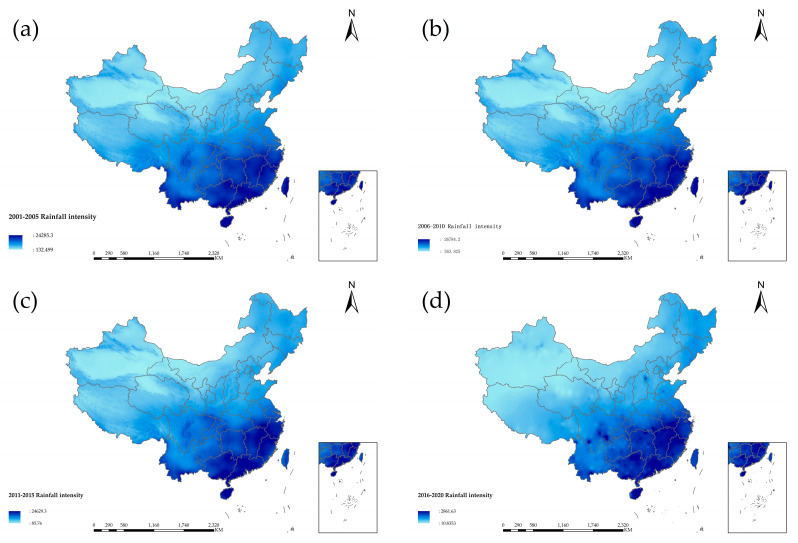
Spatiotemporal distribution of rainfall intensity in China from 2001 to 2020.

**Figure 11 entropy-24-00772-f011:**
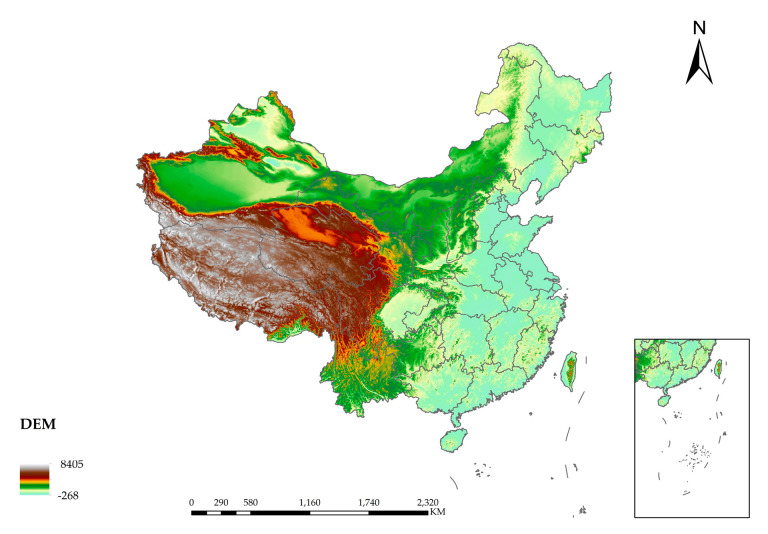
Spatial distribution map of DEM.

**Figure 12 entropy-24-00772-f012:**
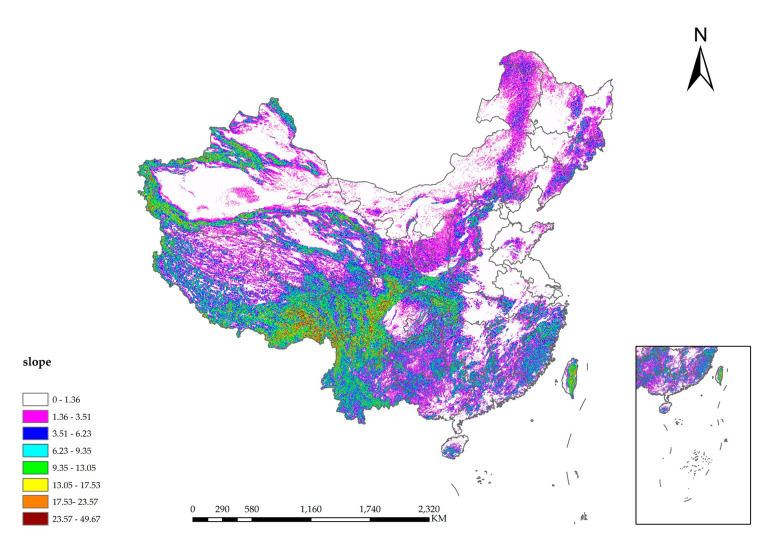
Spatial distribution map of slope.

**Figure 13 entropy-24-00772-f013:**
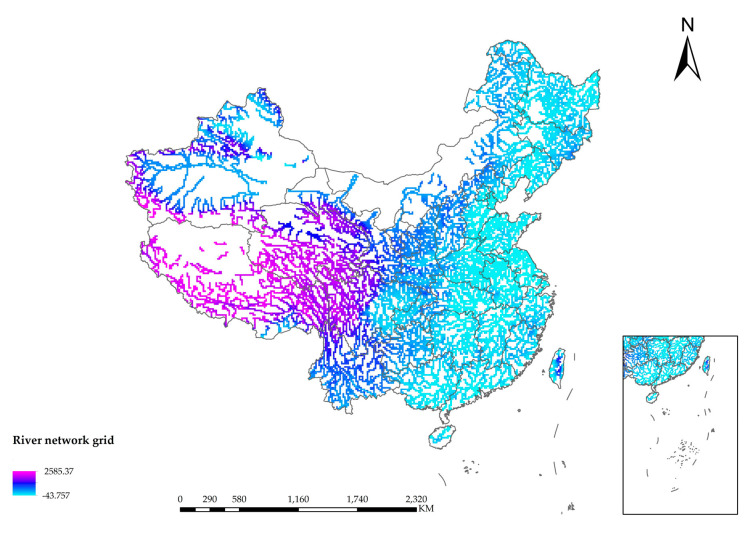
Spatial distribution map of major river networks.

**Figure 14 entropy-24-00772-f014:**
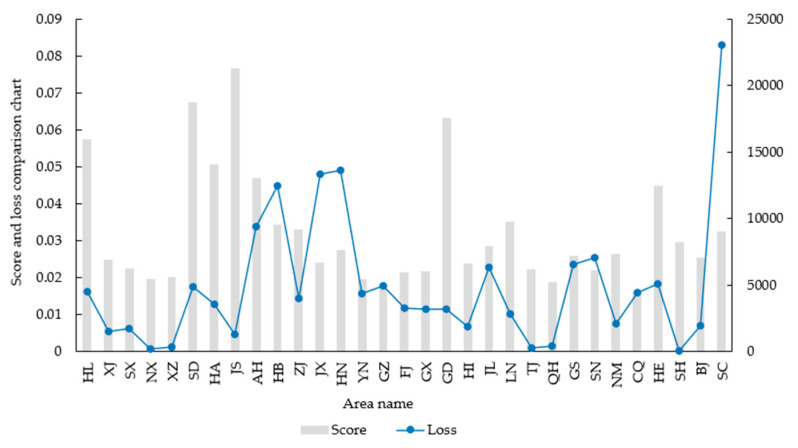
Combination graph of the final score and direct loss.

**Figure 15 entropy-24-00772-f015:**
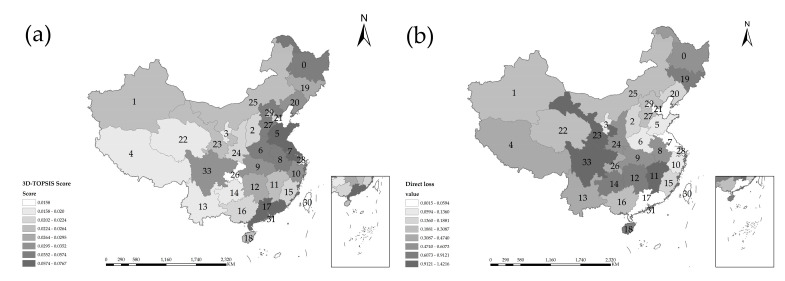
3D-TOPSIS score results in spatial distribution map.

**Figure 16 entropy-24-00772-f016:**
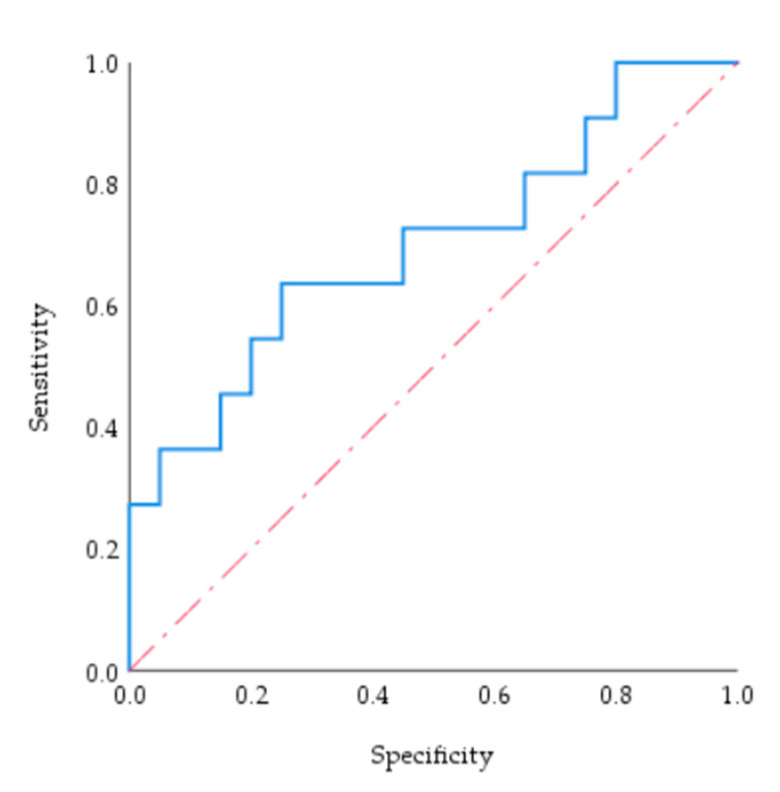
The ROC curve of the flood control capacity score to the MADM-GIS model.

**Table 1 entropy-24-00772-t001:** Abbreviations for every Chinese region.

Number	Area Name	Abbreviation	Number	Area Name	Abbreviation
0	Heilongjiang	HL	16	Guangxi	GX
1	Xinjiang	XJ	17	Guangdong	GD
2	Shanxi	SX	18	Hainan	HI
3	Ningxia	NX	19	Jilin	JL
4	Tibet	XZ	20	Liaoning	LN
5	Shandong	SD	21	Tianjin	TJ
6	Henan	HA	22	Qinghai	QH
7	Jiangsu	JS	23	Gansu	GS
8	Anhui	AH	24	Shaanxi	SN
9	Hubei	HB	25	Inner Mongolia	NM
10	Zhejiang	ZJ	26	Chongqing	CQ
11	Jiangxi	JX	27	Hebei	HE
12	Hunan	HN	28	Shanghai	SH
13	Yunnan	YN	29	Beijing	BJ
14	Guizhou	GZ	33	Sichuan	SC
15	Fujian	FJ			

**Table 2 entropy-24-00772-t002:** List of factor abbreviations and acronyms.

Abbreviation	Parameters	Abbreviation	Parameters
OIF	The objective and inclusive factor	U	Urban green area
		C	Channel density
SPF	The subjective preventive factor	W	Waterlogging prevention area
		P	Population density
FMP	The flood mitigation prominence	M	Medical institution
		G	GDP
FHD	The flood hazard degree	R	Rainfall intensity
		D	DEM
		S	Slope

**Table 3 entropy-24-00772-t003:** Key math symbols and their meanings.

Symbol	Description
i	The number of evaluation objects
j	The number of evaluation indicators
P	The forward matrix
$a_{ij}$	The elements in a forward matrix
F	Probability matrix
$F_{ij}$	Corresponding probability of $a_{ij}$
$S_{ij}$	Information entropy
$m_{ij}$	Information utility value
$W_{j}$	Weight of each indicator

**Table 4 entropy-24-00772-t004:** Key math symbols and their meanings.

Symbol	Description
$\dot{q_{ij}}$	Indicator raw data
$q_{ij}$	The index data after forwarding
$T_{ij}$	Normalized indicator data
*Z*	Weighted normalization matrix
$\omega_{ij}$	Indicator corresponding weight
$Q^{+}$	$\mathrm{Maximum}\;\mathrm{value}\;\mathrm{in}\;q_{ij}$
$Q^{-}$	$\mathrm{Minimum}\;\mathrm{value}\;\mathrm{in}\;q_{ij}$
$D_{i}{}^{+}$	$\mathrm{The}\;\mathrm{distance}\;\mathrm{between}\;\mathrm{the}\;\mathrm{evaluation}\;\mathrm{object}\;\mathrm{and}\;Q^{+}$
$D_{i}{}^{-}$	$\mathrm{The}\;\mathrm{distance}\;\mathrm{between}\;\mathrm{the}\;\mathrm{evaluation}\;\mathrm{object}\;\mathrm{and}\;Q^{-}$
$S_{i}$	Relative proximity

## Data Availability

http://www.stats.gov.cn/tjsj/ndsj/, https://www.gddat.cn/newGlobalWeb/#/chinaDisasterDatabase, https://www.resdc.cn/ (accessed on 12 March 2022).

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
