# Peer review of "Spatial-Temporal Sensitivity Analysis of Flood Control Capability in China Based on MADM-GIS Model"

_entropy, 2022, doi:10.3390/e24060772_

Round 1

Reviewer 1 Report

Dear Authors,

After reviewing your article, I have some recommendations that I would like to share with you.

  1. Please check the article for mistakes and wrong English formulation.
  2. Please observe the formating of the citations in the text. The brackets are smaller than the text.References are not in the correct template.
  3. From the introduction, it is not clear how other scientists approached the topic, in other parts of the world, so the Introduction must be extended to emphasize this aspect.
  4. The flow chart must be moved to Methodology, together with the lines 75-86. Instead, the novelty of your approach must be emphasized in the last paragraph of Introduction.
  5. Please reformulate the phrase to be clear; ''. In this paper, combined with the evaluation method of the index system, the MADM model is relatively new, the efficient and accurate assessment is completed at the same time.'' 
  6. line 154 - please give some references for MCDM. Who introduced the model? Is it implemented in software?
  7. lines 195-202 should be moved to Results and Discussion.
  8. lines 216-231 should be moved to Discussion.
  9. line 233, what do you mean by ''divided into four stages''?
  10. Analysis and Validation should be at Results and Discussion. You used here only 5 years. Where are the rest?
  11. Taylor and ROC diagrams should be firstly presented in Methodology.
  12. The article should be completely rewritten, taking into account to separate the methodology and results, and discussions. 

Reviewer 2 Report

In the paper, a flood defense capacity evaluation model is proposed. The model includes an evaluation index system, which is described in the methodology. But the paper needs a major revision, including the responses to the questions and suggestions written below;

  1. What is the DEM source and its accuracy? It should be given.
  2. It should be stated what each letter in the equation means.
  3. In the data resource part, it is mentioned that raster calculation is performed on the same spatial resolution. The spatial resolution should be given.
  4. What is the source of urban green space maps? It should be mentioned.
  5. In the figures, the mentioned cities are not possible to be recognized. It is suggested to add numbers for each polygon on the image and provide a legend showing the cities list.
  6. Put the metrics for figures 11a and 11b.
  7. In all figures, two decimal digits will be enough for the values.
  8. The manuscript does not include a deep discussion of the results. These results should be discussed with previous national and international studies to indicate its achievement.

Round 2

Reviewer 1 Report

Dear Authors,

I really appreciate the work performed to improve the manuscript. Still, some corrections should be done.

  1. Please provide the reference to Lin, Lin et al. at page 2.
  2. Please remove “ at the beginning of the first row, at page 3.
  3. On page 4, please remove the dot in (1.) and diminish the dimensions of the letters for FCS and the equation (1). The same for some letters on page 9.
  4. Page 16: Please provide the statistical tests that validate the models in (15).
  5. References are not written according to the Instructions for authors.

Reviewer 2 Report

The authors considered the suggestions and improved the manuscript, including additional information. I would recommend the revised manuscript for publication.

Round 3

Reviewer 1 Report

Dear authors,

After reading the new version of the manuscript, I noticed that it was improved. Still, there are some issues to be fixed.

1. Please leave a space before the number of the reference, For example, in ArcGIS platform[41-43], put a space after platform. It should be done all over the text.

2. In table 1, instead of serial number it should be only number.

 3. The following text Figure 4, 8&9 should be replaced by Figures 4, 8 and 9.

4. References are not in the correct format. The names of the journals maust be abbreviated.

5. In Figure 15, the loss is very high for nine zones. How does it affect the result? It does not seem that there is a good optimization....

6. The models' validation, as I asked in the previous review, has not been performed.

7. You must extend the discussions on the results by emphasizing the drawbacks and possibly imrovements.

Round 4

Reviewer 1 Report

Dear Authors,

Thank you for anserwing my questions.

The article can be published as it is. 

A link with the journals' names abbreviations is the following.

https://www.library.caltech.edu/journal-title-abbreviations

Good luck!